# Surviving the Storm: Cytokine Biosignature in SARS-CoV-2 Severity Prediction

**DOI:** 10.3390/vaccines10040614

**Published:** 2022-04-14

**Authors:** Rahnuma Ahmad, Mainul Haque

**Affiliations:** 1Department of Physiology, Medical College for Women and Hospital, Plot No 4 Road 8/9, Sector-1, Dhaka 1230, Bangladesh; rahnuma.ahmad@gmail.com; 2Unit of Pharmacology, Faculty of Medicine and Defence Health, Universiti Pertahanan Nasional Malaysia (National Defence University of Malaysia), Kem Perdana Sungai Besi, Kuala Lumpur 57000, Malaysia

**Keywords:** SARS-CoV-2 virus, pathogenesis, biomarkers, cytokines, inflammation, severe illness, cytokine storm, ARDS, multiorgan failure

## Abstract

**Simple Summary:**

The world has been stricken mentally, physically, and economically by the COVID-19 virus. However, while SARS-CoV-2 viral infection results in mild flu-like symptoms in most patients, a number of those infected develop severe illness. These patients require hospitalization and intensive care. The severe disease can spiral downwards with eventual severe damage to the lungs and failure of multiple organs, leading to the individual’s demise. It is necessary to identify those who are developing a severe form of illness to provide early management. Therefore, it is crucial to learn about the mechanisms and chemical mediators that lead to critical conditions in SARS-CoV-2 infection. This paper reviews studies regarding the individual chemical mediators, pathways, and means that contribute to worsening health conditions in SARS-CoV-2 infection.

**Abstract:**

A significant part of the world population has been affected by the devastating SARS-CoV-2 infection. It has deleterious effects on mental and physical health and global economic conditions. Evidence suggests that the pathogenesis of SARS-CoV-2 infection may result in immunopathology such as neutrophilia, lymphopenia, decreased response of type I interferon, monocyte, and macrophage dysregulation. Even though most individuals infected with the SARS-CoV-2 virus suffer mild symptoms similar to flu, severe illness develops in some cases, including dysfunction of multiple organs. Excessive production of different inflammatory cytokines leads to a cytokine storm in COVID-19 infection. The large quantities of inflammatory cytokines trigger several inflammation pathways through tissue cell and immune cell receptors. Such mechanisms eventually lead to complications such as acute respiratory distress syndrome, intravascular coagulation, capillary leak syndrome, failure of multiple organs, and, in severe cases, death. Thus, to devise an effective management plan for SARS-CoV-2 infection, it is necessary to comprehend the start and pathways of signaling for the SARS-CoV-2 infection-induced cytokine storm. This article discusses the current findings of SARS-CoV-2 related to immunopathology, the different paths of signaling and other cytokines that result in a cytokine storm, and biomarkers that can act as early signs of warning for severe illness. A detailed understanding of the cytokine storm may aid in the development of effective means for controlling the disease’s immunopathology. In addition, noting the biomarkers and pathophysiology of severe SARS-CoV-2 infection as early warning signs can help prevent severe complications.

## 1. Introduction

Severe acute respiratory syndrome coronavirus 2 (SARS-CoV-2) infections have resulted in the coronavirus disease 2019 (COVID-19) pandemic. This virus has been linked to a high morbidity and mortality rate representing an extraordinary global health challenge [1]. The concerning variants of SARS-CoV-2, which have a higher transmission rate and mortality rate, are those found in India (B.1.617., Delta), Brazil (P.1, Gamma), South Africa (B.1.351), and the UK (B.1.1.7, Alpha) [2]. A fifth concerning variant marked by WHO has emerged in South Africa and is named Omicron (B.1.1.529) [3]. Other new variants such as B.1.429, B.1.427, Epsilon (in California); B.1.525, Eta (in Nigeria); B.1.526, Iota (in New York) and Delta, B.1.617.2, B.1.617.1, and Kappa (in India) exhibit high transmissibility and decreased neutralization by post-vaccination and convalescent sera [4].

Mild to moderate respiratory system illnesses such as dry cough, difficulty in breathing, and fever are symptoms of most patients infected with SARS-CoV-2. Many infected suffer acute respiratory distress syndrome (ARDS), linked to a high mortality rate [5]. Infected individuals may also develop bronchiolitis, alveolitis, fluid and mucus accumulation, and inflammation with immune cells infiltration in the interstitium of the lung [6,7]. The target organ for SARS-CoV-2 depends on angiotensin-converting enzyme II (ACE2), which is the receptor protein targeted by the virus. High ACE 2 expression is found in the lung, heart, small intestine, kidney, and immune system [8,9,10,11]. There may be a requirement for intensive care in severe disease since there is a risk of multiple organ damage and shock [10].

The COVID-19 virus belongs to the beta coronavirus genus with single-stranded RNA. The S protein of the virus binds to the ACE2 [12]. Following this binding, the virus is transformed into a spring-like structure when the head of the S protein is cleaved by host proteases, cathepsin L transmembrane serine protease 2. The virus can then fuse with the host membrane and enter directly into the host cell through the cell surface or by endocytosis [13,14]. The viral RNA translation occurs in the host cell with immediate induction of innate immunity response through chemokine, interferon, cytokines such as interleukin 6, interleukin 1β, Granulocyte-macrophage colony-stimulating factor, and tumor necrosis factor [9,15,16].

Once infected by the SARS-CoV-2 virus, cytokine levels may increase, resulting in severe damage of tissue and, in some cases, cytokine release syndrome [17]. Several studies have noted that very high levels of inflammatory cytokines in COVID-19 link the cytokine storm with poor infection outcome [18,19,20]. The cytokine storm develops rapidly with pro-inflammatory cytokines’ overproduction and excessive immune cell activation, resulting in persistent fever, muscle pain, hypotension and ARDS, DIC, capillary leak syndrome, failure of multiple organs, and even death [21].

Cytokine storm was used as a term in the case of the graft-vs-host disease in 1993 [22] and then in organ transplantation and inflammatory conditions such as autoimmune diseases and SARS-CoV-2 infection [23,24,25,26,27,28,29].

## 2. Objectives of the Study

This study objective was to review studies that note the SARS-CoV-2 immunopathological characteristics, biomarkers, cytokines, and chemokines related to severe SARS-CoV-2 infection and mechanisms leading to cytokine storm induced by the virus. Discussion on the source, function, and signaling pathways of the individual cytokines is also part of the objectives of this study.

## 3. Material and Methods

This narrative review identifies the various mechanisms, cytokines, and immune responses involved in severe COVID-19 infection. The study was carried out between November 2021 and February 2022. The search was carried out from an electronic database using Google search engine, Google Scholar, Science Direct, PubMed, Embase, and MEDLINE. Related articles from a list of references were searched to obtain more papers. Keywords used in the search for related articles were ‘SARS-CoV-2 virus’, ‘Cytokine storm’, ‘Severe COVID-19 infection’, ‘Immune response in COVID-19 infection’, ‘Inflammatory cytokine’, ‘signaling pathways of inflammation’, ‘Inflammation in SARS-CoV-2’, and ‘Complications of severe SARS-CoV-2 infection’. The study included articles and literature dated before 2000 and articles unavailable in English. Relevant articles were hand-searched before inclusion in this study.

## 4. Immune Response in SARS-CoV-2 Infection

The immune system comprises two principal segments: the innate and the adaptive immune system. These two arrangements perform collaboratively and deal with diverse physiological functions [30,31,32]. The job of the innate immune system is to act as the first line of the host’s protective mechanism to counter any infectious microbes and harmful or sterile offending substances, including SARS-CoV-2 [33,34,35]. Innate immune physiology halts viral entry, translation, replication, and assembly [36,37]. Additionally, it assists in recognizing and eradicating infected cells and synchronizing and enhancing adaptive immunity [38,39]. The innate system also mediates these offenses through cell surface, endosomal, and cytosolic pattern recognition receptors (PRRs) and responds by means of recognizing pathogen-associated molecular patterns (PAMPs) via initiating inflammatory retaliation and programmed cell death that arrests the viral infection and encourages removal from the system [35,40,41]. Adaptive immunity is a group of specific second-line defense immune responses that are broader and more finely tuned which arise after contact with microbial antigen or a vaccination [31,42,43]. Adaptive immunity has conventionally been well-thought-out as an exclusive characteristic of craniate physiology [44], thereby controlling most viral infections [45,46]. The three principal integrants of adaptive immunity are B cells, CD4+ T cells, and CD8+ T cells [45,47,48]. B cells produce antibodies. CD4+ T cells control a spectrum of helper and effector physiological defensive capabilities. CD8+ T cells ensure death of infected cells [45,49,50].

COVID-19 infection forces dampening or detainment of the innate immune system [51,52]. The strategies espoused by SARS-CoV-2 easily evade natural antiviral defensive armaments. Consequently, it allows the virus to reproduce at full tilt and exhibit an acute increase in inflammatory response, especially persuading a vigorous type I/III interferon response, synthesis of proinflammatory cytokines, and conscription of neutrophils myeloid cells [53,54,55]. Furthermore, acute COVID-19 infection reduces broad immune cells, which comprise T, natural killer, monocyte, and dendritic cells (DCs) [56]. These cells involve innate and adaptive immune physiological issues [57]. Moreover, it was reported that DCs were pointedly turned down with physiologically poor function, and ratios of regular DCs to plasmacytoid DCs were maximized among severely acute cases of COVID-19 [56,58,59]. Additionally, it was reported that DC-SIGN gene expression was remarkably diminished in lung DCs and, in extremely severe cases, increased the number of immature DCs (CD22+ or ANXA1+ DCs) with major histocompatibility complex class II molecules (MHCII) downregulation observed because of intense deformation of DC’s physiological property [58]. Multiple studies revealed an alteration of B-cell receptors (IGHV3–23 and IGHV3–7) among SARS-CoV-2 patients, and the number of naïve B cells is reduced, although peripheral blood plasma cells were bizarrely augmented [60,61]. The main immunopathological characteristics in SARS-CoV-2 infection include monocyte and macrophage dysregulation, lymphopenia, neutrophilia, delayed or decreased type I interferon response, and cytokine storm (Figure 1) [22].

## 5. Neutrophilia

In SARS-CoV-2 infection, neutrophilia has been observed. Under physiological conditions, neutrophils phagocytose pathogens and thus protect against disease [62]. On the other hand, when there is overactivation of neutrophils, they can cause the dissolving of connective tissue and damage other cells [63].

It was noted in a clinical study that neutrophil count was higher in severe SARS-CoV-2 infection when compared to healthy subjects [64]. In a clinical trial in Wuhan, China, the neutrophil count increased in SARS-CoV-2 infected subjects who did not survive compared to those who survived. The neutrophils continued to rise until the demise of the subjects who did not survive the infection [65]. A study of single-cell sequencing [66] and another flow cytometry study [67] noted neutrophil precursors at different stages of peripheral blood mononuclear cells development of SARS-CoV-2 patients with Acute Respiratory Distress Syndrome. A clinical study that performed RNA sequencing with peripheral blood mononuclear cells observed the presence of dysfunctional mature neutrophils expressing programmed death-ligand 1 and immature neutrophil progenitors in case of severe SARS-CoV-2 infected patients [68].

Even though the possible mechanism used by the virus to cause neutrophilia in SARS-CoV-2 infection is not understood, it has been noted in a study that glycolysis regulators, pyruvate kinase M2 level and phosphorylated pyruvate kinase M2, were increased in neutrophils of SARS-CoV-2 infected subjects who were admitted to the ICU in comparison with COVID-19 patients that were not in the ICU [69,70]. This finding suggests the reprogramming of neutrophils in severe cases of SARS-CoV-2 infection leading to overproduction and overactivation of neutrophils in severe SARS-CoV-2 infection [22].

To suppress virus particles, neutrophils liberate histone-wrapped nucleic acid structures that appear like a web known as Neutrophil Extracellular Traps (NETs) [71]. NETs’ overproduction causes lung tissue injury resulting from the effects of enzymes related to NETosis such as Neutrophil elastase and Myeloperoxidase [72]. The extent of damage within the lung and the gravity of disease have been associated with the uncontrolled production of NET. Markers of NETosis are linked to lung inflammation [73]. ’Primed’ neutrophils producing NETs have been found in subjects with ARDS with pneumonia [74]. In severe SARS-CoV-2 infection, lung inflammation is a significant complication of the respiratory system that can be life-threatening [75]. A study conducted to note the damaging role of NETs in severe SARS-CoV-2 infected patients observed a rise in NETs in plasma and aspirate from the trachea in COVID-19 patients; the neutrophils of these subjects released significantly higher concentrations of NETs, and NETs were found in specimens of lung tissue from autopsies of infected subjects. NETs produced by neutrophils in COVID-19 patients were found to cause cell death in lung epithelium [76]. An investigation performed with SARS-CoV-2 infected patients’ sera noted increased citrullinated histone, myeloperoxidase DNA, and cell-free DNA. SARS-CoV-2 patients’ sera also prompted control neutrophils to release NETs [77]. NETosis triggered by a virus acts as a double-edged sword, since NETs act as a DNA web to efficiently trap viruses [78] and are also responsible for very high inflammatory processes that damage the body [79]. Symptoms of COVID-19 could thus be influenced by the link between neutrophil function in destroying virus and NETs overproduction, leading to cytokine storm [80].

## 6. Lymphopenia

A common finding in SARS-CoV-2 infected subjects related closely to the severity of the illness is Lymphopenia [8,65,81,82]. Studies have observed a reduction in percentage and count of all types of lymphocytes, including CD4+, CD8+ cytotoxic T, B cells, and natural killer cells [81,82,83,84,85,86,87,88]. Flow cytometry, nonhuman primate models of COVID-19, and single-cell sequence studies have also observed lymphopenia [66,67,89]. T cells express raised levels of exhaustion markers such as programmed cell death protein-1 mucin domain-3 and T cell immunoglobulin [84,86]. Mechanisms that may be involved in SARS-CoV-2 induced lymphocytopenia include (a) SARS-CoV-2 virus attacking T cells utilizing ACE2 receptor present on T cells, leading to the death of T cells [90,91,92], (b) destruction of lymph nodes, spleen, and secondary lymphoid tissues, and (c) cytokines promoting exhaustion and decline of T cells [81,93,94,95].

## 7. Antibody-Mediated Effect of B Lymphocytes

Immunological remembrance is a mechanism to safeguard us from reinfection [96,97]. Neutralizing antibodies (NAbs) synthesized by B cells are fundamental for the immunological defense approach and lie beneath almost all viral disease control and vaccine success [98,99]. Among COVID-19 infected patients, B cells produce antibodies that target the ACE2 receptors, which impede the entry of the SARS-CoV-2 virus into cells [100,101]. Another study revealed that mild COVID-19 infection persuades vigorous antigen-specific, long-lived humoral immune memory in humans [102]. Multiple research studies revealed that convalescent COVID-19 cases possess anti-SARS-CoV-2 antibodies and had low rates of reinfection [103,104,105]. A positive association has been observed between COVID-19 disease severity and high antibodies or B cells levels [66,106,107]. The fragment crystallizable region (Fc region) is the end area of an antibody that intermingles with cell surface receptors, termed Fc receptors (FcRs), and a number of proteins of the complement system [108]. FcRs and complement interaction augment neutralization, removal of infected cells, opsonization of virions, and modulate innate and adaptive immune activity [109,110,111]. Additionally, the Fc domain of antibodies when bound with viral proteins on the surface of virus-infected cells has been reported. This interaction promoted the release of cytotoxic substances such as perforins and granzymes, subsequently killing virus and virus-infected cells [112,113,114]. A similar observation was noticed in COVID-19 cases [113,114] and other viral diseases [110,113,115,116,117,118,119,120].

## 8. Monocyte and Macrophage Dysregulation

A significant role in the innate immune response for viral infection and inflammation is played by monocytes and macrophages. A study where single-cell sequencing of RNA was carried out noted a significant reduction in CD16^+^ nonclassical monocytes and CD14^+^ CD16^+^ intermediate monocytes, but a marked rise in CD14^+^ classical monocytes in the blood of SARS-CoV-2 infected subjects having severe symptoms. Nonclassical monocytes maintain the homeostasis of blood vessels and anti-inflammatory function, while the classical monocytes can convert to tissue macrophages leading to the inflammatory response [121]. A high expression of chemokines and cytokines by monocyte and macrophage in broncho-alveolar lavage fluid suggests inflammatory response [122]. Another study observed a shift from CD16^+^ to CD14^+^ in SARS-CoV-2 patients with ARDS and reduced CD16+ monocyte in peripheral blood in these subjects. It was also noted that gene encoding chemokines and cytokines were not upregulated in monocytes in the periphery, suggesting that cytokine storm progression in SARS-CoV-2 patients may not involve monocytes in peripheral blood [66]. The study of the broncho-alveolar lavage fluid subpopulation of leukocytes found that activation markers such as HLA-DR, CD69, CD 64, and CD16 were greater in macrophages in broncho-alveolar lavage fluid than in macrophages in the periphery [35]. These studies suggest that monocytes and macrophages in the respiratory system produce inflammatory chemokines and cytokines in severe SARS-CoV-2 infection and thus participate in the cytokine storm [22].

## 9. Response of Interferon Type 1

Clearance of the virus and regulation of innate and adaptive immunity involves the response of Interferon type 1, which is vital for combating virus infection [123]. Cells of natural immune response recognize SARS-CoV-2 RNA through NOD-like receptors, retinoic acid-inducible gene-I-like receptor, and toll-like receptor [124] with eventual activation of Interferon regulatory factor 3/7, activator protein 1, and nuclear factor kβ and promoting Interferon type 1 and inflammatory cytokines synthesis. There is the activation of tyrosine kinase 2 or Janus Kinase–signal transducer and activator of transcription 1/2 (STAT1/2) pathway and initiation of Interferon stimulated genes by Interferon type 1 [125,126,127]. However, studies suggest a marked reduction in the Interferon type 1 protective response in severe SARS-CoV-2 infection [84,128,129]. Interferon type 1 response may be inhibited through M protein, ORF6a protein, ORF3a protein, and N proteins, which are structural components of the SARS-CoV-2 virus [130,131,132,133]. Moreover, NSP1 protein found in the SARS-CoV-2 virus may cause inhibition of transcription of ISG and phosphorylation of STAT1. Inhibition of Interferon type 1 may also result from reduced plasmacytoid dendritic cells, which is a producer of Interferon type 1 during viral infection [134,135]. Plasmacytoid dendritic cells have been noted to be decreased in SARS-CoV-2 infected patients’ blood and more so in those suffering severe infection [66,84]. However, a study has observed delayed Interferon type 1 response in those patients who are critically ill, since their broncho-alveolar lavage fluid contained Interferon type 1 and ISG despite the low level of systemic Interferon type 1 [127,134]. A delayed Interferon type 1 response aggravates inflammation and hinders virus clearance [135,136].

## 10. Interleukin 6

Several studies have found an increased level of Interleukin 6 in individuals suffering from severe SARS-CoV-2 infection [126,137,138,139,140]. In Germany, a study found that Interleukin 6 levels greater than 80 pg/mL along with CRP greater than 97 mg/L showed a high degree of sensitivity and specificity for predicting respiratory failure [138]. Another retrospective study in China with 150 individuals observed a rise in the level of Interleukin 6 in severe SARS-CoV-2 subjects [137]. Interleukin 6 (released by dendritic cells, monocytes, and macrophages) is a notable activator of the pathway of JAK/STAT3, and the axis of Interleukin 6, Janus Activated Kinase, and STAT3 has a close relationship with SARS-CoV-2 severity [141,142]. JAK/STAT3 pathway activation leads to increased T helper 17 differentiation, B cells, CD^+^ 8 T cells, neutrophil migration, and decreased Tregs formation, leading to further Interleukin 6 formation promoting inflammation [22,143].

Over-activation of the Interleukin 6–Janus Activated Kinase–STAT3 pathway may produce Interleukin 6, monocyte chemoattractant protein 1, vascular endothelial growth factor, and Interleukin 8 and decrease E Cadherin on endothelial cells [144]. Interleukin 6 may cause acute phase proteins production such as hepcidin, thrombopoietin, fibrinogen, CRP, ferritin, and complement C3 [145,146]. Studies have found monocyte chemoattractant protein 1 may be responsible for adhesion molecule formation, Vascular Smooth muscle cells proliferation, and thermogenesis [147,148,149,150]. Raised vascular endothelial growth factor and lowered E cadherin may result in increased permeability and leakage of the blood vessel and therefore promote dysfunction in the lung during cytokine storm in SARS-CoV-2 infection [151].

Studies have found that COVID-19 virus may cause angiotensin II expression [152,153]. Binding of the angiotensin II with angiotensin II type 1 receptor leads to JAK/STAT3 pathway activation, increasing Interleukin 6 production [154,155]. Thus, it is possible that there is an enhancement of Interleukin 6 by the SARS-CoV-2 virus and this therefore acts as a biochemical signature in severe SARS-CoV-2 infection [156].

## 11. Interleukin 7

T lymphocyte maintenance, survival, and differentiation require Interleukin 7 [157,158,159,160]. It is also essential for developing lymphoid tissue, innate lymphoid cell maintenance, and development [22]. Raised levels of Interleukin 7 in SARS-CoV-2 infection and an association between increased Interleukin 7 with the severity of disease have been observed in recent studies [19,111,161].

## 12. Interleukin 10

T helper cell 2, CD 8^+^ T cells, dendritic cells, B cells, Macrophages, Tregs, and natural killer cells release Interleukin 10. This Interleukin 10 has immune regulatory functions. This cytokine signaling occurs through the JAK/STAT 3 pathway. Interleukin 10 can cause activation of mast cells and promote CD8^+^ T cells, natural killer cells, and B cells. This cytokine can enhance Treg development and limit the immune function of macrophage and dendritic cells [162]. Although the enhanced production of Interleukin 10 attempts to subdue the hyperactive immune system, the role of Interleukin 10 is insufficient in the cytokine storm in SARS-CoV-2 infection, where there is an immense production of inflammatory cells [163].

A study observed a significantly higher level of Interleukin 10 in the plasma of SARS-CoV-2 infected patients who took admission into the ICU compared to those who were not admitted [161]. The findings of a clinical trial have noted Interleukin 10 to be a possible disease severity indicator since the study observed that serum levels of Interleukin 10 were significantly increased in patients who were critically ill than the patients suffering moderate to severe infection, with the positive correlation between Interleukin 10 and serum C Reactive Protein [164]. Another follow-up study that consisted of 71 SARS-CoV-2 infected patients and 18 control subjects found that Interleukin 10 production in the early disease stage had a significant correlation with severity of disease [165]. Interleukin 10 may cause exhaustion of T lymphocytes in the early stage of COVID-19 infection, damaging the infected patient. Thus, using a neutralizing antibody for Interleukin 10 blocking during the early stages of the COVID-19 infection can be of therapeutic use [163].

## 13. Interleukin 12

B cells, macrophage, and dendritic cells release Interleukin 12, which promotes T helper cell 1 and T helper cell 17 proliferation; enhances natural killer cells’ cytotoxic function, and increases the release of Interferon γ, natural killer cells, macrophages, dendritic cells, and T helper 1 cells [166,167]. Cytokine storm aggravates several immune cells’ activation. However, a recent study found no difference between plasma Interleukin 12 levels between SARS-CoV-2 infected patients and healthy individuals [161].

## 14. Interleukin 2

CD4^+^ T cells produce Interleukin 2, which has a fine-tuning effect on the immune response and has a vital role in natural killer cells, CD8^+^ cells, and CD4^+^cells differentiation and expansion utilizing the Interleukin 2–Janus Activated Kinase–STAT 5 pathway of signaling [168,169,170]. A study performed in China with 54 SARS-CoV-2 infected subjects noted decreased plasma Interleukin 2 level in subjects suffering critical condition (*n* = 6) compared to subjects with severe illness (*n* = 14). Additionally, in comparison with healthy subjects, patients with critical and severe disease had significantly reduced Interleukin 2 Rα in peripheral blood mononuclear cells. In severe SARS-CoV-2 infection, the researchers suggested decreased Interleukin 2, Interleukin 2R, Janus Activated Kinase 1, and STAT5 results in lymphopenia [171].

## 15. Interleukin 17

T helper 17, ILC3, and CD8^+^ produce Interleukin 17 and take part in various processes of inflammation [172,173]. This cytokine’s function (which may protect against infection but can also have a harmful inflammatory effect) depends on the location and type of tissue in which this cytokine is produced [174]. A study observed that Interleukin 17 played a significant part in the hyperactivation of immune cells and caused dysfunction of target organs in COVID-induced cytokine storms. The cytokine promoted neutrophil recruitment and caused inflammation infiltration, tissue remodeling, and fever [175].

## 16. Tumor Necrosis Factor α

Tumor Necrosis Factorα (TNFα) is an inflammatory cytokine produced mainly by T cells, macrophages, and monocytes [176,177]. TNFα causes activation of Nuclear Factor κβ and leads to activation of NF κβ signaling. This results in inflammatory gene expression via the TNFR1 receptor [178,179,180]. TNFα and NF κβ together enhance systemic inflammation and promote epithelial cell apoptosis. Thus, it is vital in initiating and hyper-activating immune cells in cytokine storms [22]. TNFα has been observed to be raised in severely ill COVID-19 patients [19,93,161,181]. Table 1 includes studies performed on COVID-19 infected patients showing the immune response of the body to SARS-CoV-2 virus.

## 17. Interferon γ

Interferon γ (IF γ) is produced by T cells, NK cells, and macrophages. It takes part in processes of inflammation. It causes activation of T cells, NK cells, and macrophages and acts via JAK activation [184,185]. IF γ has significant involvement in the cytokine storm [186]. Various studies have found a rise in IF γ levels in SARS-CoV-2 infected subjects [161,187]. However, another study found a reduced level of IF γ in severely ill COVID-19 patients compared to those with mild infection. It may have resulted from T cells’ exhaustion in severe SARS-CoV-2 infection [188].

## 18. Interleukin 1β

Interleukin 1β is produced by macrophages and promotes immune cell migration to the site of inflammation, cytokine, and adhesion factor release, and activation of the NFκβ pathway and differentiation of T helper cell 17 [189,190]. Mature IL1β is formed by cleaving of inactive IL1β. This cleaving is achieved by a complex formed by NLRP3 proteins with NLRP3 inflammasomes (caspase 1 and ASC) [22]. Studies have noted that IL1β may play an important role in the coronavirus cytokine storm [191,192,193]. Huang et al. found a rise in the level of IL1β in SARS-CoV-2 infected patients [161]. Another study conducted by Zhang et al. noted the increase in the level of IL1β in severely ill SARS-CoV-2 infected patients [194]. Activation of NLRP 3 may result from excessive reactive oxygen species production and inactive IL1β cleaving and worsening COVID-19 cytokine storm [22,195,196,197].

## 19. Granulocyte-Macrophage-Colony-Stimulating Factor (GM-CSF) Signaling

Hematopoietic cells, endothelial cells, epithelial cells, and other cells produce the cytokine called GM-CSF [198]. GM-CSF promotes alveolar macrophage homeostasis in low levels under normal conditions [22,199]. However, during a cytokine storm, GM-CSF triggers myelopoiesis which causes myeloid cells’ aggregation at the inflammation site and accelerates inflammation reactions [200]. A recent study observed a rise in GM-CSF levels in mild and severe cases of SARS-CoV-2 infection [161].

## 20. C Reactive Protein (CRP)

CRP (an acute-phase protein) increases during the early stages of inflammation [201] and has been noted to rise in patients suffering from SARS-CoV-2 infection [202,203,204]. COVID-19 virus causes macrophage complement system activation resulting in hyperinflammatory conditions, and thus may raise the CRP level [205,206]. Cytokine, like TNFα and Interleukin 6, during inflammatory stages, causes stimulation of hepatocytes to release CRP [202]. Studies have noted the rise in CRP to be significantly related to SARS-CoV-2 induced cytokine storm in severely ill subjects [206,207]. Stringer et al. reported that mortality from COVID-19 correlated with CRP cut-off value > 40 mg/L and suggested that this finding may be used as a guideline by clinicians for appropriate management and planning for care in advanced stages of the disease [207].

Xie et al., in a study carried out in 2020 with 140 hospitalized SARS-CoV-2 infected subjects having moderate to severe bronchopneumonia and needing oxygen supplementation, observed that subjects with oxygen saturation ≤90% had a median CRP level of 76.5 mg/L, which was significantly higher than subjects having oxygen saturation of >90%, that is, median CRP level 12.7 mg/L. Such findings suggest severe involvement of lung increase CRP levels [208]. A similar association between CRP levels with the early stage of COVID-19 was also noted by Wang [209].

## 21. D Dimer

D dimer is a breakdown compound of fibrin formed by fibrinolysis induced by plasmin. It is a biomarker for disorders of thrombosis [210,211]. Studies have observed an association between mortality among hospitalized COVID-19 patients and a rise in D dimer (>1.5–2.0 ug/mL) [212,213]. Yao found, in 75% of the hospitalized patients, 184 subjects out of 248 in Wuhan, China, the elevation of D dimer. The rise in D dimer was significantly associated with the severity of the disease [213]. D dimer measurement is of prognostic value for thrombosis since the thrombotic condition in arteries and veins has been found in COVID-19 patients [212,214].

## 22. Cytokine Storm

Cytokines such as TNF α, interleukin-1,2,6,7,8, Interferon γ, granulocyte-macrophage colony-stimulating factor, monocyte chemoattractant protein 1, and granulocyte colony-stimulating factor are components that contribute to cytokine storm in SARS-CoV-2 infection [67,70,84,181,215,216,217,218,219]. COVID-19 induced cytokine storm is more aggressive than other cytokine storms, as more cytokines and lymphopenia are involved, which is less common in other cytokine storms. Innate immune cells are mainly involved in SARS-CoV-2 induced cytokine storm [220]. Once the SARS-CoV-2 virus enters the cell, the viral RNA is sensed by PRRs as a pathogen-associated molecule pattern (PAMP) in the cell. There is then activation of NFκβ and IRF3/7, which leads to the formation of Interferon I (IFNI) and inflammatory cytokines [221,222,223]. SARS-CoV-2 virus removes the protective response of IFN I by several mechanisms. There is the enhancement of overwhelming inflammation and impairment of virus clearance as there are massive amounts of inflammatory cytokine production. The expected protective host immune response converts to harmful inflammation in SARS-CoV-2 infection [21].

The respiratory epithelium is infected by the SARS-CoV-2 virus, leading to chemokines such as CCL 2 and CCl3.CCl5, CXCL10, and pro-inflammatory cytokines such as Interleukins 1, 6, 8, 12, and TNF α. These chemokines and cytokines promote the aggregation of innate immune cells such as neutrophils, dendritic cells, natural killer cells, macrophages, and monocytes. There is also activation of adaptive immune cells such as CD8^+^ and CD4^+^ cells, which cause the continued formation of pro-inflammatory cytokines such as Interferon γ, Interleukin 2, and TNF α. These events cause induction of granulopoiesis and myelopoiesis and promote further damage to the lung and epithelium [22,224]. Systemic cytokine overproduction aggravates activation of macrophages as well as phagocytosis of erythrocytes, resulting in anemia [225,226]. There is also disturbance of hemostasis and coagulation, which leads to thrombosis and capillary leak syndrome [227,228]. Such a sequence of events eventually results in Acute Respiratory Distress Syndrome, failure of multiple organs, and demise (Figure 2) [21]. Although immune regulatory cells such as Tregs produce Interleukin 10 and Tumor Growth factor β that can reduce overactive immune response, it is insufficient to antagonize the hyper inflammation state produced in cytokine storm [22,229,230].

A study advised the inclusion of a ratio of peripheral blood oxygen saturation to a fraction of inspired oxygen, cytokine to chemokine ratio, neutrophil to lymphocyte ratio, C reactive protein, and ferritin for diagnosis of cytokine storm in SARS-CoV-2 infection [231]. In another study, certain criteria have been suggested to diagnose SARS-CoV-2 induced cytokine storm. The criteria include (a) Lymphocyte < 10.2%, absolute count of neutrophil > 11.4 × 10^3^/mL, albumin < 2.87 mg/mL, (b) D-dimer > 4930 ng/mL, troponin > 1.09 ng/mL, lactate dehydrogenase > 416 U/L, alanine aminotransferase > 60 IU/L, aspartate aminotransferase > 87 IU/L, and (c) blood urea nitrogen:creatinine ratio > 29, potassium > 4.9 mmol/L, chloride > 106 mmol/L, and anion gap < 6.8 mmol/L. Also included are C-reactive protein > 4.6 mg/dL and ferritin > 250 ng/mL 182 [232]. Hyperinflammation screening in the laboratory and performing an HS score in severe COVID-19 illness have been proposed in a study for identification of SARS-CoV-2 induced cytokine storm [19].

## 23. Predictive Factors and High-Risk Case of Cytokine Strom

Multiple studies reported that elevated liver enzymes, LDH, D-dimers, and troponin I indicate massive cell death, which leads to significant systemic tissue damage, particularly in the liver, the cardiovascular system, and kidney, suggestive of the COVID-19 cytokine storm (COVID-CS) [233,234,235,236]. Another study reported that five markers (Alanine aminotransferase (ALT), aspartate aminotransferase (AST), D-dimers, lactate dehydrogenase (LDH), and troponin I) of tissue injury were more significantly raised among patients with COVID-CS than among other SARS-CoV-2 cases [182,237,238,239]. COVID-19 patients’ ALT and AST levels exceed three times the upper limit, indicating liver damage [240]. Exalted D-dimer status is a forecaster for severe SARS-CoV2-infection and indicates thromboembolic disorders. Nevertheless, it is also a direct result of acute lung injury, as observed in COVID-19 pneumonia [241,242,243]. It has been reported that elevated LDH levels among COVID-19 cases suggest acute cell death and severe cases [244,245]. It has been revealed that, among COVID-19 cases, the LDH level rising ~6 and ~16-times in odds denotes developing severe disease and mortality, respectively [244]. Cardiac troponin is considered a marker of myocardial or heart muscle injury, and is thereby well-known as the first emblem of heart damage in blood. Several earlier cases of COVID-19 in Wuhan, China were hospitalized for respiratory issues and had a high level of troponin [246]. Raised troponin levels are frequently found among SARS-CoV-2 cases and suggestively indicate deadly consequences [247,248]. One more systematic meta-analysis analyzing 40 papers reported that interleukin-6, ferritin, leukocytes, neutrophils, lymphocytes, platelets, C-Reactive Protein, procalcitonin, LDH, AST, creatinine, and D-dimer are authoritative biomarkers of COVID-CS. Higher-up levels of interleukin-6 and hyperferritinemia need to be considered as a warning of fatal clinical outcomes, including death [249].

The factors that promote the risk of COVID-CS are male, >40 years, the positive test result for replicative SARS-CoV-2 RNA, absolute lymphocyte count (<0.72 × 109/L), and dynamics in the National Early Warning Score 2 (NEWS2) score, as well as LDH (>23 pg/mL), D-dimer, ferritin, serum CRP (>50 mg/L), and IL-6 levels [250]. One more meta-analysis verified that the inflammatory biomarkers, notably white blood count (WBC), absolute lymphocyte count (ALC), absolute neutrophil count (ANC), platelet count (PLT), C-reactive protein (CRP), ferritin, D-dimer, LDH, fibrinogen, and erythrocyte sedimentation rate (ESR), were correlated with the prognosis and severity among COVID-19 pediatric cases presenting with the multisystem inflammatory syndrome (MIS) [251]. One Mexican study reported that the mortality rate was 36.84% of the studied sample. The patients who passed away were 59.71 ± 13.83 years, statistically significantly higher than those who survived (43.29 ± 11.80 years). Additionally, serum levels of Interleukin 6 (IL-6) were significantly higher in patients with fatal outcomes. Furthermore, a correlation was detected between IL-6 levels with lymphocyte count, LDH, CRP, and procalcitonin (PCT) [252]. One more study reported that during admission of COVID-19 patients, the levels of SpO2, lymphocyte, CRP, PCT, and LDH often determine the prognosis of severity and clinical outcome. Additionally, systematic inflammation with cardiac complications frequently regulates fatality in severe SARS-CoV-2 apart from acute respiratory distress syndrome [253]. Another study determined that IL-6 and CRP levels serve as robust interpreters for those patients who require ventilator support [136]. One more study reported that the median levels of IL-6 were <1.5 pg/mL {(Interquartile range (IQR) < 1.50–2.15]), 1.85 pg/mL (IQR < 1.50–5.21), and 21.55 pg/mL (IQR 6.47–94.66) for the common, severe, and critical SAR-CoV-2 groups, respectively (*p* < 0.001). The follow-up kinetics revealed serum IL-6 was high in critical patients, even when cured. An IL-6 concentration higher than 37.65 pg/mL was predictive of in-hospital death Area under Curve (AUC) 0.97 (95% CI 0.95–0.99), *p* < 0.001 with a sensitivity of 91.7% and a specificity of 95.7% [254]. Multiple studies report the poor clinical outcome of COVID-19 patients frequently observed among the elderly population and in the presence of comorbidities which include *angiotensin-converting enzyme-2* polymorphism, cancer, cancer chemotherapy, chronic kidney disease, thyroid disorder, diabetes, CVD, hypertension, oxidative stress, vitamin deficiency, or hematological disorders [255,256,257,258]. Another study reported that at least 50% of patients who develop COVID-CS had been elderly and suffering from hypertension or diabetes mellitus or both [259]. The aging process and the mentioned comorbidities promote oxidative stress and endothelial dysfunction [260,261]. These two comorbidities frequently act as determinant factors of SAR-CoV-2 disease prognosis, severity and even death [259].

## 24. The Complications of Cytokine Storm

The principal cause of COVID-CS cases worsens, often leading to death because of the hyper-responsiveness of the immune system and mortality of COVID-19 patients [185,262,263]. Additionally, multiple research studies revealed that gravely sick SARS-CoV-2 patients frequently encountered severe hypercoagulable states that increased the negative impact on survival possibility [214,264,265,266,267,268]. This tendency to build thrombosis has been identified by the noticeable upsurge in D-dimer in almost all hospital-admitted SARS-CoV-2 patients. D-dimer is the end product of fibrinolysis and is found in plasma or whole blood. Raised D-dimer denotes the active fibrinolysis and ongoing coagulation [210,213,269,270,271,272,273,274,275,276,277,278,279].

It has been reported that COVID-CS often initiates augmented myocardial oxygen consumption, endothelial dysfunction, coronary spasm, coronary artery disease, myocardial injury, repressed cardiac function, arrhythmias, microthrombi, heart, and circulatory failure [137,151,272,273,274,275,276,277]. Another study reported that COVID-CS could cause damage to the kidney because of sepsis, shock, hypoxia, and rhabdomyolysis [278,279,280]. Renal injury with subsequent development of hematuria, and proteinuria was detected [281]. It has been reported that acute hemorrhagic necrotizing encephalopathy is related to COVID-CS [282]. Another study reported that COVID-CS caused damage to the blood–brain barrier (BBB) and later instigated brain necrosis [283].

The raised ALT levels denote hepatic deteriorated function and are frequently correlated with COVID-19 infection [240,284,285]. Kumar et al. conducted a systematic review and reported that the cumulative prevalence of acute hepatic damage was projected at 23.7 (16.1–33.1) per 100 patients with SARS-CoV-2 [286]. Mao et al. conducted another systematic review and meta-analysis that reported 19% (range: 1–53%) of COVID-19 cases had the hepatic injury. The hypoalbuminemia (26.3–30.9 g/L) was observed with severe cases of SARS-CoV-2 infection. Correspondingly, the pooled prevalence of raised liver enzymes for ALT, AST, and total bilirubin was found to be 18% (13–25%), 21% (14–29%), and 6% (3–11%), respectively [287]. However, the detailed pathology of hepatic damage due to SARS-CoV-2 infection is poorly understood [288,289]. Nevertheless, it has been postulated that hepatic injury is possibly due to direct cytopathic effects of SARS-CoV-2, impaired immune function, and COVID-CS correlated with multiorgan failure, hypoxia-reperfusion related damage, and idiosyncratic drug-induced hepatic impairment due to medicines utilized in the treatment of COVID-19 [288,290,291,292]. There are reports of ophthalmic involvement with SARS-CoV-2 infection, which include retinovascular disease, uveitis, optic neuropathies, dry eye, foreign body sensation, itching, blurring of vision, redness, tearing, itching, eye pain, conjunctival discharge, foreign body sensation, and orbital fungal co-infections [293,294,295]. Elderly SARS-CoV-2 infected cases with high fever, increased neutrophil/lymphocyte ratio, and high levels of acute-phase reactants appeared as the precipitating factors for ophthalmic complications [296].

## 25. The Current Therapeutic Options for Cytokine Storm and Time to Intervene

The recent clinical data exhibit that tocilizumab (TCZ), an anti-interleukin-6, is reasonably effective with a low potential to cause adverse drug effects (ADRs) and, in doing, results in a lower death rate among SARS-CoV-2 infected hospitalized patients [297,298,299]. Another study reported that TCZ’s pharmacotherapeutic benefit was more observed when administered along with corticosteroids (CS) and when this combination was given within the first 10 days of symptoms [300,301], whereas one more study reported that TCZ as monotherapy showed substantial benefit among patients who had a comparatively low level of the ratio of ferritin/CRP, thereby quickly decreasing the CRP, IL-6, IFN-γ, IP-10, and MCP-1 levels. Nevertheless, the high proportion of ferritin/CRP is related to the swift deterioration of respiratory function. This study suggested that for patients with a high ratio of ferritin/CRP, TCZ in combination with glucocorticoids is a better therapeutic option [302].

SARS-CoV-2 infection dysregulated cytokine release, or COVID-CS, is the major pathogenesis and remains the primary reason for multiple organ failure and death [303,304]. The role of glucocorticoids comes to the forefront, especially in managing the hyperinflammatory state of SARS-CoV-2, otherwise called COVID-CS, because of its dominant role in immunosuppression [305]. Nevertheless, the timing of CS administration remains very critical in SARS-CoV-2 infection [305,306]. Additionally, proper CS and dose adjustment selection remains ambiguous [307]. If CS administered in COVID-CS achieves a beneficial effect, it can be detrimental if provided too early in COVID-19 cases [305]. After that, glucocorticoids should be justified based on two diverse pathophysiological phases of the SAR-CoV-2 infection: (1) in the initial stage, administration of pharmacological doses of CS might essentially upsurge the plasma viral load because of immunosuppression, and (2) in the late phase (hyperinflammatory COVID-CS stage), CS pharmacodynamic properties will suppress hyperinflammation and alleviate the COVID-19-induced cytokine storm [308]. Furthermore, adrenal cortical steroid, programmed cell death protein (PD)-1/PD-L1 checkpoint inhibition, cytokine-adsorption devices, intravenous immunoglobulin, monoclonal antibodies, low molecular weight heparin, and antimalarial agents have been reported as potentially beneficial and dependable therapeutic options for the management of COVID-CS [297,309,310,311,312,313,314,315,316,317,318,319,320]. Plausibly, these therapeutic options possibly will work and safeguard COVID-CS patients. Nevertheless, the probability of successful treatment greatly depends on selecting the correct patients, correct medication, appropriate dose, and correct time to intervene [321,322,323].

## 26. Limitations of this Paper

The following were some limitations of the study:

This study is a narrative review, so a meta-analysis was not conducted.

Studies in languages other than English could not be included.

Articles that need to be accessed through institutional access could not be accessed.

## 27. Conclusions

Studies of clinical and basic research have noted that the characteristics of SARS-CoV-2 induced cytokine storm and have facilitated our knowledge of the pathophysiology of SARS-CoV-2 infection and cytokine storm. There is the involvement of many more cytokines in the COVID-19 infection-related cytokine storm than in the case of other conditions. The cytokine storm, in this case, is far more damaging and inflammation is much more severe. It has been noted that the innate immune response produces these inflammatory cytokines. The role of each cytokine in the development of cytokine storms is not yet completely understood. However, there is undoubtedly a close association between pathogenic alterations in SARS-CoV-2 infection and the SARS-CoV-2 induced cytokine storm. Induction of prolonged and excessive cytokines by SARS-CoV-2 may result in lung damage and failure of multiple organs. Studies have been performed on peripheral blood chemokines and cytokines, and the immunopathological damage due to the SARS-CoV-2-induced cytokine storm may be the reason behind severe complications and death in COVID-19 infection. There is a need for extensive molecular and epidemiological studies to understand the specific inflammatory signaling pathways and develop effective means for curtailing the virus from further spreading globally.

## 28. Recommendation

We recommend extensive molecular and epidemiological studies should be carried out in the future to understand the inflammatory mediators’ roles further. Treatment strategies targeting the signaling pathways and cytokines of inflammation should be developed. Drugs that target cytokines play a vital role in the cytokine storm, and cells that produce cytokines, such as macrophages, monocytes, dendritic cells, and natural killer cells, need to be developed. Studying the immune-regulatory system and recognizing anti-inflammatory cytokines such as Interleukin 37’s effects on inhibiting inflammation can help develop therapeutics to improve patient condition [232,324,325]. In addition, specific treatment plans need to be prepared, as the extent of the cytokine storm and treatment may vary with comorbidity, age, and immunity status. Early detection of severe SARS-CoV-2 infection biomarkers may help prevent multiorgan damage and death. Therefore, the development of precise treatment plans and increasing knowledge regarding the pathogenesis of COVID-19 infection is consequently of great importance.

## 29. Article Highlights

A significant part of the world population has been affected by the devastating SARS-CoV-2 infection. Evidence suggests that the pathogenesis of SARS-CoV-2 infection may result in immunopathology such as neutrophilia, lymphopenia, decreased response of type I interferon, monocyte, and macrophage dysregulation. Excessive production of different inflammatory cytokines leads to cytokine storm in COVID-19 infection. The large quantities of inflammatory cytokines trigger several inflammation pathways through tissue cell and immune cell receptors. Such mechanisms eventually lead to complications such as acute respiratory distress syndrome, intravascular coagulation, capillary leak syndrome, failure of multiple organs, and, in severe cases, death. Treatment strategies targeting the signaling pathways and cytokines of inflammation should be developed. Early detection of severe SARS-CoV-2 infection biomarkers may help prevent multiorgan damage and death. Therefore, developing precise treatment plans and increasing knowledge regarding the pathogenesis of COVID-19 infection is of great importance.

## Figures and Tables

**Figure 1 vaccines-10-00614-f001:**
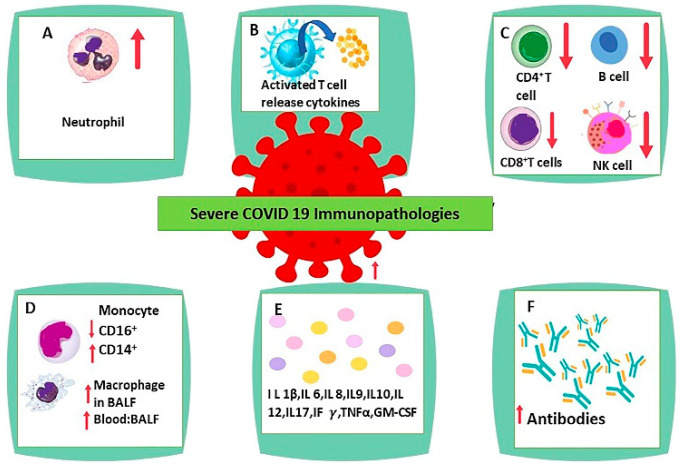
Immunopathological characteristics in severe SARS-CoV-2 infection include: (**A**) Neutrophilia, (**B**) T cell activation with cytokine release, (**C**) Lymphopenia, (**D**) Monocyte and macrophage dysregulation, (**E**) Cytokine Storm, and (**F**) Increased production of antibodies. Note: Upward arrow mean increased and downward arrow mean decreased.

**Figure 2 vaccines-10-00614-f002:**
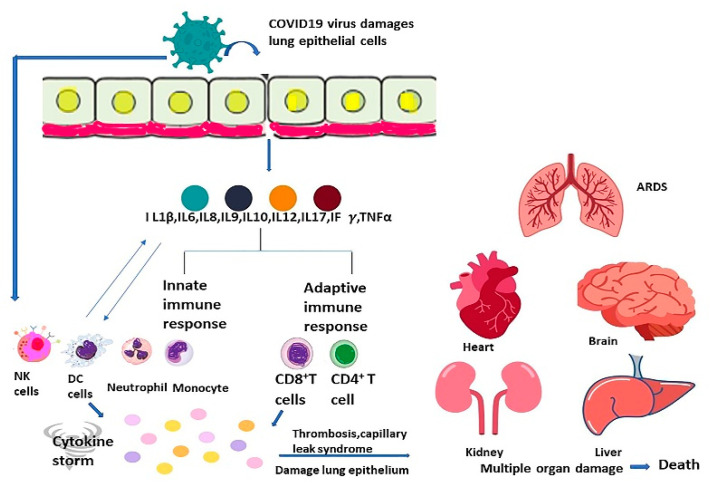
The virus infects the host and can then fuse with the host membrane and enter directly into the host cell through the cell surface or by endocytosis. Then, there is the immediate induction of immune response through chemokine, interferon, cytokines such as interleukin 6, interleukin 1β, Interleukin 8, Interleukin 9, Interleukin 10, Ifγ, and tumor necrosis factor. The cytokine levels may increase, resulting in severe damage of tissue and cytokine storm. Cytokine storm develops rapidly with pro-inflammatory cytokines’ overproduction and excessive immune cell activation resulting in ARDS, DIC, capillary leak syndrome, failure of multiple organs, and even death. Notes: Arrow denotes follow-up consequences and double arrow mean interchangeable situation.

**Table 1 vaccines-10-00614-t001:** Illustrating the studies showing immune response of human body to SARS-CoV-2 infection [65,66,67,82,84,138,139,182,183].

Reference	Study Population	Study Design	Study Period	Subgroup	Results
Masso-Silva et al. 2022 [183]	N = 31	Case series	11 days	Case:16 Critically ill COVID-19 patients with APACHE(Acute Physiology and Chronic Health Evaluation) II scores of 7–27 on intensive care unit (ICU) admission. Control:15 healthy subjects	Plasma cytokine profiles and complete blood counts of COVID-19 patient demonstrated elevations in IL-8, IL-6, neutrophil:lymphocyte ratio (mean, 9.3). Profiling of specific cytokines relevant to neutrophil activity showed broad elevations across IP-10, IL-6, IL-8, granulocyte macrophage colony-stimulating factor (GM-CSF), interleukin 1β, interleukin 10, and tumor necrosis factor alpha (TNF-α) in the circulation of critically ill COVID-19 patients both early in their hospitalization and were remained raised throughout their hospitalization, measured at multiple time points
Wang et al. 2020 [65]	N = 138 Age= 56 years(median age)	Case series	1 month (1 January–3 February 2020)	102 (73.9%) were admitted to isolation wards, and 36 (26.1%) were admitted and transferred to the ICU because of development of dysfunction of organ	Common symptoms included fever (136 [98.6%]), fatigue (96 [69.6%]), and dry cough (82 [59.4%]). Lymphopenia (lymphocyte count, 0.8 × 10^9^/L [interquartile range 0.6–1.1]) in 97 patients (70.3%), Raised Neutrophil count in 36 ICU patients 4.6 (2.6–7.9) *p* = <0.001 and elevated lactate dehydrogenase (261 U/L [IQR, 182–403]) in 55 patients (39.9%). Chest computed tomographic scans revealed bilateral patchy shadows or ground glass opacity in the lungs of all patients
Wilk et al. 2020 [66]	N = 13 age ≥18 years	Cross sectional study	2–3 weeks	single-cell RNA sequencing (scRNA-seq) to profile peripheral blood mononuclear cells (PBMCs) was done. Case: 7 patients hospitalized for COVID-19, 4 of whom had acute respiratory distress syndrome Control: 6 healthy controls.	HLA class II downregulation was noted and a developing neutrophil population were observed that appears closely related to plasmablasts appearing in patients with acute respiratory failure requiring mechanical ventilation.
Ronit et al. 2021 [67]	N = 4 Age = 40–75 years	Cross sectional study	2months 21 days	SARS-CoV-2 patients confirmed by PCR, with presence of ARDS determined according to the Berlin criteria and less than 72 h of mechanical ventilation after admittance to the intensive care unit (ICU)	Immature neutrophils were raised in both blood and lungs, whereas CD4 and CD8 T-cell lymphopenia was observed in the 2 compartments. However, regulatory T cells and T_H_17 cells were found in higher fractions in the lung. Lung CD4 and CD8 T cells and macrophages expressed an even higher upregulation of activation markers than in blood. Cytokines were expressed at high levels both in the blood and in the lungs, most markedly, IL-1RA, IL-6, IL-8, IP-10, and monocyte chemoattactant protein-1, pointing to hyperinflammation.
Wang et al. (2020) [82]	N = 60 Age = 60 years(medan)	Cross sectional study	5 weeks	Levels of peripheral lymphocyte subsets were measured by flow cytometry in 60 hospitalized COVID-19 patients before and after treatment	Total lymphocytes, CD4^+^ T cells, CD8^+^ T cells, NK cells and B cells reduced in COVID-19 patients, and severe cases had a lower level than mild cases. Lymphocyte subsets showed a significant relation with inflammatory state in COVID-19, especially CD4^+^/CD8^+^ ratio and CD8^+^ T cells. Following treatment, clinical response was observed in 37 patients (67%), with an rise in CD8^+^ T cells and B cells
Hadjadj et al. (2020) [84]	N = 68	Cross sectional study	10 days	COVID 19 patient = 50 Mild-moderate = 15 Severe = 17 Critical = 18 Healthy subjects = 18	in severe and critical patients, there was highly impaired interferon (IFN) type I response (characterized by no IFN-β and low IFN-α production and activity), which was related with a persistent viral load in blood and hyperinflammatory response. Inflammation was characterized by increased tumor necrosis factor–α and interleukin-6 production and signaling.
Herold et al. (2020) [138]	N= 89	Cohort study	5 weeks	initial evaluation cohort (n = 40) which was followed by a validation cohort that was separated temporally (n = 49)	CRP and IL 6 levels in the evaluation cohort were0.86 and 0.97, and they were similar in the validation cohort (0.83 and 0.90, respectively)
Laing et al. (2020) [139]	N = 73; Age = 61 years(median)	Cross sectional study	3 weeks	Patients with COVID-19 = 63 Control group not suffering from COVID-19 = 10	patients exhibited considerable person to person in number of variation in B cell, ranging from overt cytopenia (<10^4^ B cells ml^−1^) to atypically high counts (2–3 × 10^5^ mL^−1^). IL-6 and IL-10 levels were also highly raised in COVID-19 and the rise were related to severity.
Caricchio et al. (2021) [182]	N = 513; Age = 58.3 years	Cohort study	5 weeks	513 patients were enrolled in the cohort and considered eligible must have met the following criteria on hospital admission: (1) signs and symptoms of COVID-19 infection (fever, generalised malaise, cough and shortness of breath) up to 1 week prior to admission to hospital and (2) findings of ground-glass opacity (GGO) by high-resolution CT (HRCT) of the chest as per radiology reading	Elevated levels of IL6 was observed in most COVID 19 patients which was higher significantly in COVID-CS (35 vs. 96 pg/mL) confirming strong inflammation. The white blood cells, and particularly neutrophils and monocytes, were significantly increased in the COVID-CS group, suggesting innate immunity has a active role in Cytokine storm. The lymphocytes were decreased, on average half of the lower limit of normal value, indicating a functional depletion of the adaptive immunity

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
