# Peer review of "Surviving the Storm: Cytokine Biosignature in SARS-CoV-2 Severity Prediction"

_vaccines, 2022, doi:10.3390/vaccines10040614_

Round 1
Reviewer 1 Report
This review manuscript summarizes insights from the literature on the cytokine response to SARS-CoV2 infection, including the so-called cytokine storm. The review has some merits in presenting a broad spectrum of findings. However, there are substantial weaknesses.
Several statements fail to make the right conclusions, if they are not even wrong.
Example 1:
Line 106: “The immune response in the case of COVID-19 infection includes innate immune response overactivation and adaptive immune response impairment.” This is not true in most cases. The innate and adaptive responses work well, reactive T-cells and antibodies are generated.
Example 2:
Lines 176-178: “In an animal study, the protective effect of antibodies against SARS-CoV-2 was observed[72]. Further studies are needed to develop antibody-mediated therapy and vaccine for SASR-CoV-2 infection [22].” This statement is outdated, which is obvious even to non-expert readers who are vaccinated against SARS-CoV-2 and have developed antibodies.
An important weakness of the manuscript is the fact that most citations are outdated. There are very few references to papers published in 2020 and even the year 2021 is under-represented. The summary of old papers in such a rapidly moving field does not deliver much value to the reader. It would be good to focus the manuscript on few aspects, cite new studies and delete old ones.
If the authors extend the scope of their review to paper published after february 2022, please include insights from a very recent review of the immunopathology of COVID-19 (Merad et al., Science 375, 1122–1127, 11 March 2022).
The title does not fit to the content of the manuscript. There is not much information about "prediction", actually the word "predict" is used only once in all the main text.
As this is a Review manuscript, a “Material and Methods” section is not necessary. Maybe this section of the text can remain under a different heading.
In some cases the authors used capital letters incorrectly, e.g. Dendritic cells, Natural Killer cells (line 383). Please carefully check capitalization of words throughout the manuscript.
Line 67: dunnage? Maybe “damage”
Author Response
Reviewer I
Open Review
(x) I would not like to sign my review report.
( ) I would like to sign my review report.
English language and style
( ) Extensive editing of English language and style required.
(x) Moderate English changes required.
( ) English language and style are fine/minor spell check required.
( ) I don't feel qualified to judge about the English language and style
|
Is the work a significant contribution to the field? |
|
|
Is the work well organized and comprehensively described? |
|
|
Is the work scientifically sound and not misleading? |
|
|
Are there appropriate and adequate references to related and previous work? |
|
|
Is the English used correct and readable? |
Comments and Suggestions for Authors
This review manuscript summarizes insights from the literature on the cytokine response to SARS-CoV2 infection, including the so-called cytokine storm. The review has some merits in presenting a broad spectrum of findings. However, there are substantial weaknesses.
Several statements fail to make the right conclusions, if they are not even wrong.
Example 1:
Line 106: “The immune response in the case of COVID-19 infection includes innate immune response overactivation and adaptive immune response impairment.” This is not true in most cases. The innate and adaptive responses work well, reactive T-cells and antibodies are generated.
Thanks Sir. We Have Totally Altered this Section. Line 106-144.
Example 2:
Lines 176-178: “In an animal study, the protective effect of antibodies against SARS-CoV-2 was observed[72]. Further studies are needed to develop antibody-mediated therapy and vaccine for SASR-CoV-2 infection [22].” This statement is outdated, which is obvious even to non-expert readers who are vaccinated against SARS-CoV-2 and have developed antibodies.
An important weakness of the manuscript is the fact that most citations are outdated. There are very few references to papers published in 2020 and even the year 2021 is under-represented. The summary of old papers in such a rapidly moving field does not deliver much value to the reader. It would be good to focus the manuscript on few aspects, cite new studies and delete old ones.
Thanks Sir. We altered added a number of references from 2022. Line No.: 208.
If the authors extend the scope of their review to paper published after february 2022, please include insights from a very recent review of the immunopathology of COVID-19 (Merad et al., Science 375, 1122–1127, 11 March 2022).
Thanks, Sir, for the Valuable Comments. We added this reference in our paper. Reference 59.
The title does not fit to the content of the manuscript. There is not much information about "prediction", actually the word "predict" is used only once in all the main text.
Thanks for valuable comment. We have added a new section. Line No.: 465-527.
As this is a Review manuscript, a “Material and Methods” section is not necessary. Maybe this section of the text can remain under a different heading.
Dear Sir, earlier we experienced and need to add Materials and Methods. They referred this paper https://researchintegrityjournal.biomedcentral.com/articles/10.1186/s41073-019-0064-8. Thereby, we add and desire to retain. Moreover, REVIEWER II does not have any objection.
In some cases, the authors used capital letters incorrectly, e.g. Dendritic cells, Natural Killer cells (line 383). Please carefully check capitalization of words throughout the manuscript.
Line 67: dunnage? Maybe “damage”
Sorry, Sir. We have corrected it.
Submission Date
10 March 2022
Date of this review
16 Mar 2022 15:28:27

Reviewer 2 Report
Ahmad and Haque described cytokine biosignature in SARS-CoV-2 severity prediction. The review article discusses the immune Response in SARS-CoV-2 Infection, including neutrophilia, lymphopenia, antibody-mediated effect of B Lymphocytes, monocytes-macrophage dysregulation, response of interferon type 1, and cytokines such as IL-6, IL-7, IL-10, IL-12. IL-2, IL-17, TNF-a, IFN-y, IL-1B, GM-CSF, CRP, D Dimer and cytokine storm.
There are missing points in the review that need to be included to improve the quality of this review
a) Who are the risk group patients could be affected by COVID19 cytokine storm?
b) What are the complications of cytokine storm?
c) How to reduce/ calm the cytokine storm?
d) How the decision making can be affected by the storm?
Author Response
Reviewer II
Open Review
(x) I would not like to sign my review report.
( ) I would like to sign my review report.
English language and style
( ) Extensive editing of English language and style required
( ) Moderate English changes required
(x) English language and style are fine/minor spell check required
( ) I don't feel qualified to judge about the English language and style
Is the work a significant contribution to the field?
Is the work well organized and comprehensively described?
Is the work scientifically sound and not misleading?
Are there appropriate and adequate references to related and previous work?
Is the English used correct and readable?
Comments and Suggestions for Authors
Ahmad and Haque described cytokine biosignature in SARS-CoV-2 severity prediction. The review article discusses the immune Response in SARS-CoV-2 Infection, including neutrophilia, lymphopenia, antibody-mediated effect of B Lymphocytes, monocytes-macrophage dysregulation, response of interferon type 1, and cytokines such as IL-6, IL-7, IL-10, IL-12. IL-2, IL-17, TNF-a, IFN-y, IL-1B, GM-CSF, CRP, D Dimer and cytokine storm.
There are missing points in the review that need to be included to improve the quality of this review
- a) Who are the risk group patients could be affected by COVID19 cytokine storm?
Dear Sir Thanks for kind and expert comment. We added a new section. Line No.: 465-527.
- b) What are the complications of cytokine storm?
Thanks, Sir, for kind and expert comment. We added a new section. Line No.: 527-566.
- c) How to reduce/ calm the cytokine storm?
- d) How the decision making can be affected by the storm?
Thanks, Sir, for kind and expert comment. We added a new section Line No.: 567-600 .
Submission Date
10 March 2022
Date of this review
27 Mar 2022 04:17:19
Round 2
Reviewer 1 Report
The manuscript has been improved.
Reviewer 2 Report
The revised manuscript is improved